# Design deep neural network architecture using a genetic algorithm for estimation of pile bearing capacity

**Tuan Anh Pham\*, Van Quan Tran[ID], Huong-Lan Thi Vu, Hai-Bang Ly**

University of Transport Technology, Hanoi, Vietnam

\* anhpt@utt.edu.vn

## Abstract

Determination of pile bearing capacity is essential in pile foundation design. This study focused on the use of evolutionary algorithms to optimize Deep Learning Neural Network (DLNN) algorithm to predict the bearing capacity of driven pile. For this purpose, a Genetic Algorithm (GA) was developed to select the most significant features in the raw dataset. After that, a GA-DLNN hybrid model was developed to select optimal parameters for the DLNN model, including: network algorithm, activation function for hidden neurons, number of hidden layers, and the number of neurons in each hidden layer. A database containing 472 driven pile static load test reports was used. The dataset was divided into three parts, namely the training set (60%), validation (20%) and testing set (20%) for the construction, validation and testing phases of the proposed model, respectively. Various quality assessment criteria, namely the coefficient of determination ($R^2$), Index of Agreement (IA), mean absolute error (MAE) and root mean squared error (RMSE), were used to evaluate the performance of the machine learning (ML) algorithms. The GA-DLNN hybrid model was shown to exhibit the ability to find the most optimal set of parameters for the prediction process. The results showed that the performance of the hybrid model using only the most critical features gave the highest accuracy, compared with those obtained by the hybrid model using all input variables.

## 1. Introduction

In pile foundation design, the axial pile bearing capacity ($P_u$) is considered one of the most critical parameters [1]. Throughout years of research and development, five main approaches to determine the pile bearing capacity have been adopted, namely the static analysis, dynamic analysis, dynamic testing, pile load testing, and in-situ testing [2]. It is needless to say each of the above methods possesses advantages and disadvantages. However, the pile load test is considered as one of the best methods to determine the pile bearing capacity in view of the fact that the testing process is close to the working mechanism of driven piles [3]. Having said that, this method remains time-consuming and unaffordable for small projects [3], the development of a more feasible approach is vital. Thus, many studies have been conducted to determine the

**Data Availability Statement:** All relevant data are within the paper and supporting information files.

**Funding:** The authors received no specific funding for this work.

**Competing interests:** The authors have declared that no competing interests exist.

pile bearing capacity in taking advantage of the in-situ test results [4]. Meanwhile, the European standard (Euro code 7) [5] recommends using several ground field tests such as the dynamic probing test (DP), press-in and screw-on probe test (SS), standard penetration test (SPT), pressuremeter tests (PMT), plate loading test (PLT), flat dilatometer test (DMT), field vane test (FVT), cone penetration tests with the measurement of pore pressure (CPTu). Among the above approaches, the SPT is commonly used to determine the bearing capacity of piles [6].

Many contributions in the literature relying on the SPT results have been suggested to predict the bearing capacity of piles. As examples, Meyerhof [7], Bazaraa and Kurkur [8], Robert [9], Shioi and Fukui [10], Shariatmadari *et al.* [11] have proposed several empirical formulations for determining the bearing capacity of piles in sandy ground. Besides, Lopes and Laprovitera [12], Decort [13], the Architectural Institute of Japan (AIJ) [14] have introduced several formulations to determine the pile bearing capacity for various types of soil, including sandy and clayed ground. Overall, traditional methods have used several main parameters to estimate the mechanical properties of piles, such as pile diameter, pile length, soil type, number of SPT blow counts of each soil layer. However, the choice of appropriate parameters, along with the failure in covering other parameters, have led to the disagreement of results given by these methods [15]. Therefore, the development of an universal approach for the selection of a suitable set of parameters is imperative.

Over a half-decade, a newly developed approach using machine learning (ML) algorithms has been widely used to deal with real-world problems [16], especially in civil engineering applications. Employing ML algorithms, many practical problems have been successfully addressed and thus, paved the way for many promising opportunities in the construction industry [17–26]. Moreover, miscellaneous ML algorithms have been developed, for instance, decision tree [22], hybrid artificial intelligence approaches [27–29], artificial neural network (ANN) [30–35], adaptive neuro-fuzzy inference system (ANFIS) [36,37] and support vector machine (SVM) [16], for analyzing technical problems, including the prediction of pile mechanical behavior.

It is worth noticing that the development of the artificial neural network (ANN) algorithm has gained intense attention to treat design issues in pile foundation. For example, Goh *et al.* [38,39] have presented an ANN model to predict the friction capacity of driven piles in clays, in which the algorithm was trained by on-field data records. Besides, Shahin *et al.* [40–43] have used an ANN model to predict the driven piles loading capacity and drilled shafts using a dataset containing in-situ load tests along with CTP results. Moreover, Nawari et al. [44] have presented an ANN algorithm to predict the settlement of drilled shafts based on SPT data and shaft geometry. Momeni *et al.* [45] have developed an ANN model to predict the axial bearing capacity of concrete piles using Pile Driving Analyzer (PDA) from project sites. Last but not least, Pham *et al.*[15] have also developed an ANN algorithm and Random Forest (RF) to estimate the axial bearing capacity of driven pile. Regarding other ML models, Support Vector Machine Regression (SVR) and "nature inspired" meta-heuristic algorithm, namely Particle Swarm Optimization (PSO-SVR) [46] have bene used to predict the soil shear strength. Furthermore, Pham *et al.* [47] have presented a hybrid ML model combining RF and PSO (PSO-RF) to predict the undrained shear strength of soil. Also, Momeni et al. [48] have developed an ANN-based predictive model optimized with Genetic Algorithm (GA) technique to choose the best weights and biases of ANN model in predicting the bearing capacity of piles. In addition, Hossain *et al.* [49] used GA to optimize parameters of three hidden layers deep belief neural network (DBNN), include number of epochs, number of hidden units and learning rates in the hidden layers. It is interesting to notice that all the studies have confirmed the effectiveness when implementing the hybrid ML models as a practical and efficient tool in

solving geotechnical problems, and particularly the axial bearing capacity of pile. Despite the recent successes of machine learning, this method has some limitations to keep in mind: It requires large amounts of of hand-crafted, structured training data and cannot be learned in real time. In addition, ML models still lack the ability to generalize conditions other than those encountered during the training. Therefore, the ML model only correctly predicts in a certain data range but is not generalized in all cases.

With a particular interest in a recently developed Deep Learning Neural Network (DLNN), which has gained tremendous success in many areas of application [50–54], the main objective of this study is dedicated to the development of a novel hybrid ML algorithm using DLNN and GA to predict the axial load capacity of driven piles. For this aim, a dataset consisting of 472 pile load test reports from the construction sites of Ha Nam—Vietnam was gathered. The database was then divided into the training, validation, and testing subsets, relating to the learning, validation and phases of the ML models. Next, a novel ML algorithm using GA-DLNN hybrid model was developed. ML model using GA is used to select the most important input variables to create a new smaller dataset due to the reason that many unimportant input variables could reduce the accuracy of output forecasting. Next, a GA-DLNN hybrid model was used to optimize the parameters of the DLNN model. The optimal architecture of DLNN is used to test with the new dataset and compare with the full-size case of input variables. Besides, DLNN model can be optimized to better estimate axial load capacity of pile, including number of hidden layers, number of neurons in each hidden layer, activation function for hidden layers and training algorithm. Various error criteria, especially, the mean absolute error (MAE), root mean squared error (RMSE), the coefficient of determination ($R^2$) and Index of Agreement (IA)—were applied to evaluate the prediction capability of the algorithms. In addition, 1000 simulations relating to the random shuffling of dataset were conducted for each model in order to evaluate the accuracy of final DLNN model precisely.

## 2. Significance of the research study

The numerical or experimental methods in the existing literature still have some limitations, such as lack of data set samples (Marto et al.[55] with 40 samples; Momeni et al. [45] with 36 samples; Momeni et al.[56] with 150 samples; Bagińska and Srokosz [57] with 50 samples; Teh et al. [58] with 37 samples), refinement of ML approaches or failure to fully consider key parameters which affects the predicting results of the model.

For this, the contribution of the present work can be marked through the following ideas: (i) large data set, including 472 experimental tests; (ii) reduce the input variables from 10 to 4 which help the model achieve more accurate results with faster training time, (iii) automatically design the optimal architecture for the DLNN model, all key parameters are considered, include: the number of hidden layers, the number of neurons in each hidden layer, the activation function and the training algorithm. In which, the number of hidden layers is not fixed but can be selected through cross-mating between the parent with different chromosome length. Besides, the randomness in the order of the training data set is also considered to assess the stability of predicting result of models with the training, validate and testing set.

## 3. Data collection and preparation

### 3.1. Experimental measurement of bearing capacity

The experimental database used in this study was derived from pile load test results conducted on 472 reinforced concrete piles at the test site in Ha Nam province–Vietnam (Fig 1A). In order to obtain the measurements, pre-cast square-section piles with closed tips were driven to the ground by hydraulic pile presses machine with a constant rate of penetration. The tests

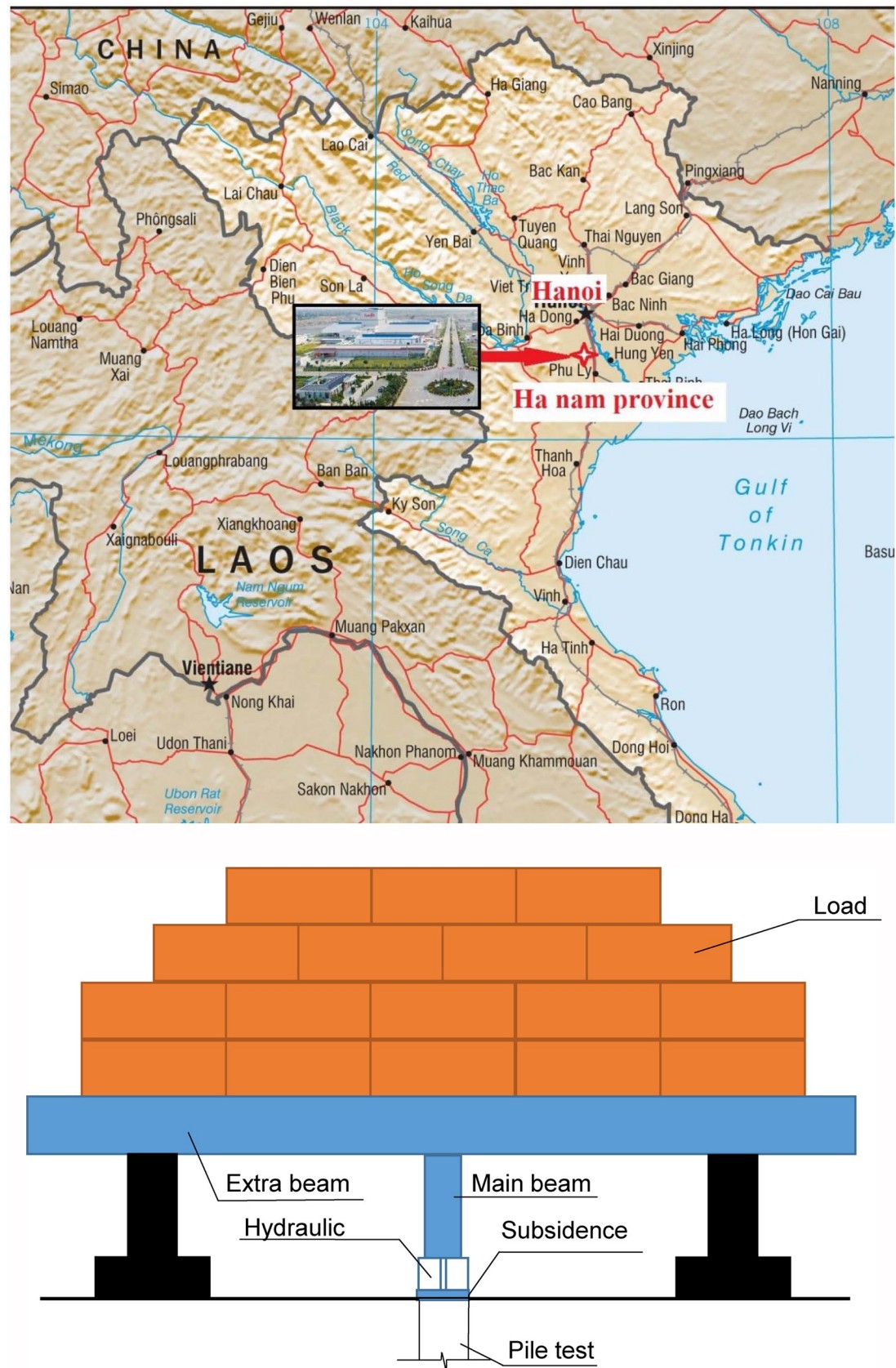

**Fig 1.** (**a**) Experimental location(*); (**b**) experimental layout. (*): Source: CIA Maps.

started at least 7 days after the piles had been driven, and the experimental layout is depicted in Fig 1B. It can be seen that the load increased gradually in each pile test. Depending on the design requirements, the load could be varied up to 200% of the pile load design. The time required to reach 100%, 150%, and 200% of the load could last for about 6 h to 12 h or 24 h, respectively. The bearing capacity of piles was determined following these two principles: (i) when the settlement of pile top at the current load level was 5 times or higher than the settlement of pile top at the previous load level, the pile bearing capacity was taken as the given failure load; (ii) when the load—settlement curve was nearly linear at the last load level, condition (i) could not be used. In this case, the pile bearing capacity was approximated as the load level when the settlement of the pile top exceeded 10% of the pile diameter.

## 3.2. Data preparation

The primary goal of the development of ML algorithms is to estimate the axial bearing capacity of the pile accurately. Therefore, as a first attempt, all the known factors affecting the pile bearing capacity were considered. Besides, it was found that most traditional approaches have used three groups of parameters: the pile geometry, pile constituent material properties, and soil properties [7–14]. It is worth noticing that the depth of the water table was not considered since it is shown that this effect have already been accounted in SPT blow counts [59]. The bearing capacity of piles was predicted based on the soil properties, determined through SPT blow counts (N) along the embedded length of the pile. In this study, the average number of SPT blows along the pile shaft ($N_{sh}$), and tip ($N_t$) was used. In addition, according to Meyerhof's recommendation (1976) [7], the average SPT ($N_t$) value for 8D above and 3D below the pile tip was also utilized, where D represented the pile diameter.

Consequently, the input variables in this work were: (1) pile diameter (D); (2) thickness of first soil layer that pile embedded ($Z_1$); (3) thickness of second soil layer that pile embedded ($Z_2$); (4) thickness of third soil layer that pile embedded ($Z_3$); (5) elevation of the natural ground ($Z_g$); (6) elevation of pile top ($Z_p$); (7) elevation of extra segment pile top ($Z_t$); (8) deepness of pile tip ($Z_m$); (9) the average SPT blow count along the pile shaft ($N_{sh}$) and (10) the average SPT blow count at the pile tip ($N_t$). The axial pile bearing capacity was considered as the single output ($P_u$). For illustration purposes, a diagram for soil stratigraphy and input, output parameters are depicted in Fig 2.

The dataset containing 472 samples is statistically introduced and summarized in Table 1, including several pile tests, min, max, average and standard deviation of the input and output variables. As showed in Table 1, the pile diameter (D) ranged from 300 to 400 mm. The thickness of the first soil layer that pile embedded ($Z_1$) ranged from 3.4 m to 5.7 m. The thickness of the second soil layer that pile embedded ($Z_2$) varied from 1.5 m to 8 m. The thickness of the third soil layer that pile embedded ($Z_3$) ranged from 0 m to 1.7 m, where a value of 0 means that the pile was not embedded in this layer. Besides, the elevation of pile top ($Z_p$) varied from 0.7 m to 3.4 m. The elevation of natural ground ($Z_g$) ranged from 3.0 m to 4.1 m. The elevation of extra segment pile top ($Z_t$) varied from 1.0 m to 7.1 m. The deepness of pile tip ($Z_m$) ranged from 8.3 m to 16.1 m. The average SPT blow count along the pile shaft ($N_{sh}$) ranged from 5.6 to 15.4. The average SPT blow count at the pile tip ($N_t$) ranged from 4.4 to 7.8. The axial bearing capacity load of pile ($P_u$), ranged from 407.2 kN to 1551 kN with a mean value of 955.3 kN, and a standard deviation of 355.4 kN. Besides, the histograms of all the input and output variables are shown in Fig 3. An example of 100 data samples is given in the appendix (S1 Appendix).

In this study, the collected dataset was divided into the training, validation, and testing datasets. The training part (60% of the total data) was used to train the ML models. The validation part (20% of the total data) was used to give an estimate of model skill and tuning model's

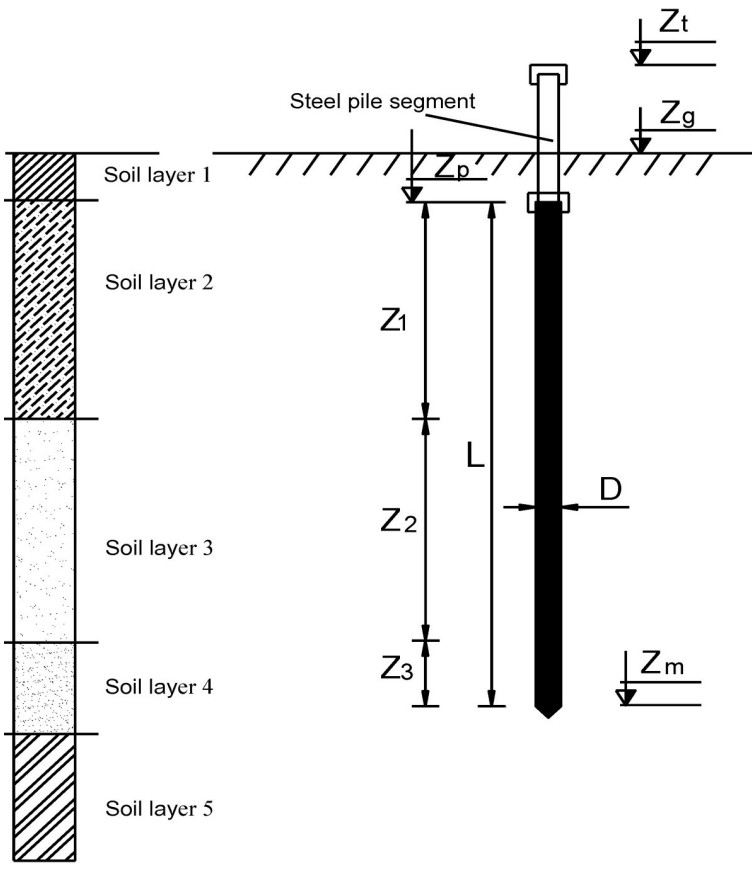

**Fig 2. Diagram for stratigraphy and pile parameters.**

**Table 1. Inputs and output of the present study.**

| N° | D | $Z_1$ | $Z_2$ | $Z_3$ | $Z_p$ | $Z_g$ | $Z_t$ | $Z_m$ | $N_{sh}$ | $N_t$ | $P_u$ |
|----|----|----|----|----|----|----|----|----|----|----|----|
| Unit | mm | m | m | m | m | m | m | m | - | - | kN |
| 1 | 400 | 4.35 | 8 | 0.95 | 2.05 | 3.41 | 2.06 | 15.35 | 13.3 | 7.6 | 1110.6 |
| 2 | 300 | 3.4 | 5.25 | 0 | 3.4 | 3.47 | 3.42 | 12.05 | 8.65 | 6.75 | 610.7 |
| 3 | 400 | 4.35 | 8 | 1.06 | 2.05 | 3.56 | 2.1 | 15.46 | 13.41 | 7.66 | 1224.8 |
| . | . | . | . | . | . | . | . | . | . | . | . |
| . | . | . | . | . | . | . | . | . | . | . | . |
| . | . | . | . | . | . | . | . | . | . | . | . |
| 470 | 300 | 3.4 | 5.2 | 0 | 3.4 | 3.43 | 3.43 | 12 | 8.6 | 6.73 | 585.35 |
| 471 | 400 | 3.45 | 8 | 0.19 | 2.95 | 3.56 | 2.97 | 14.59 | 11.64 | 7.52 | 1318 |
| 472 | 400 | 3.45 | 8 | 0.27 | 2.95 | 3.63 | 2.96 | 14.67 | 11.72 | 7.57 | 1152 |
| Min | 300.0 | 3.4 | 1.5 | 0.0 | 0.7 | 3.0 | 1.0 | 8.3 | 5.6 | 4.4 | 407.2 |
| Average | 359.4 | 3.8 | 6.5 | 0.3 | 2.9 | 3.5 | 3.0 | 13.4 | 10.5 | 7.0 | 955.3 |
| Max | 400.0 | 5.7 | 8.0 | 1.7 | 3.4 | 4.1 | 7.1 | 16.1 | 15.4 | 7.8 | 1551.0 |
| SD | 49.2 | 0.5 | 1.6 | 0.4 | 0.6 | 0.1 | 0.6 | 1.8 | 2.2 | 0.6 | 355.4 |

SD = Standard deviation.

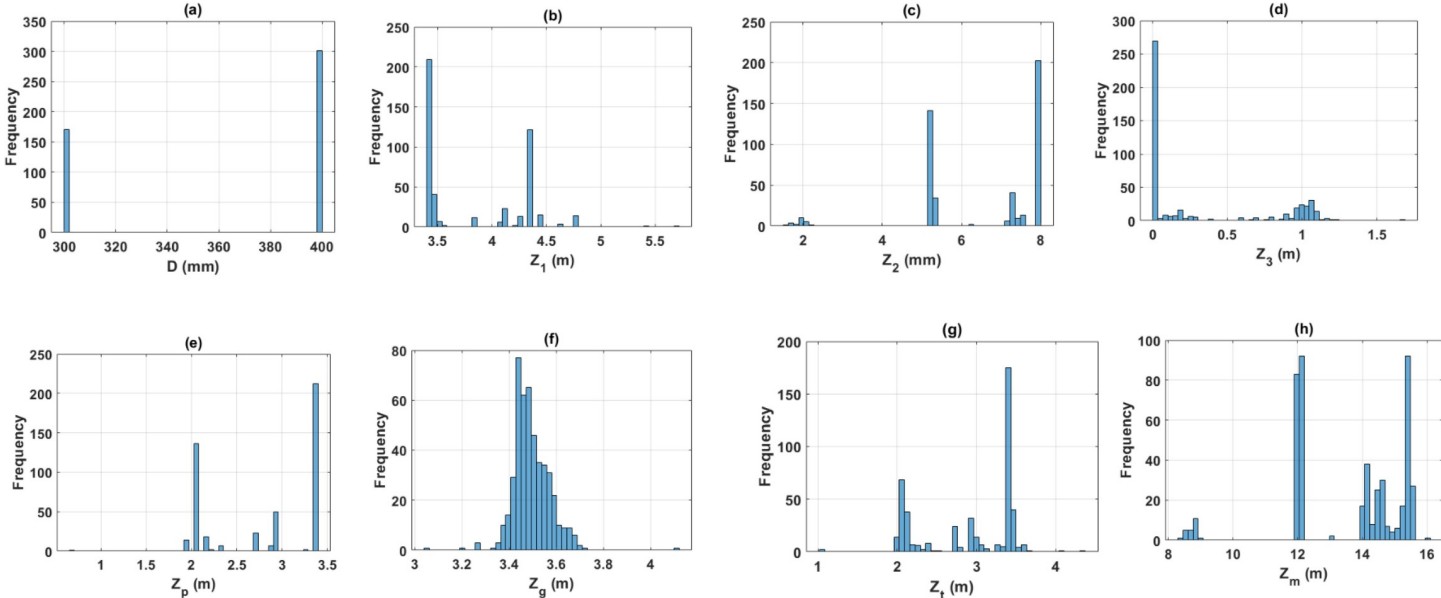

**Fig 3. Histograms of the variables used in this study.**

hyperparameters whereas testing data (20% of the remaining data), which was unknown during the training and validation phases, was used to validate the performance of the ML models.

## 4. Machine learning methods

### 4.1. Deep learning neural network (DLNN) with multi-layer perceptron

The multi-layer perceptron (MLP) is a kind of feedforward artificial neural network [60]. In general, the MLP includes at least three units, called the layers: the input layer, the hidden layer, and the output layer. When the hidden layer consists of more than two layers, the multi-

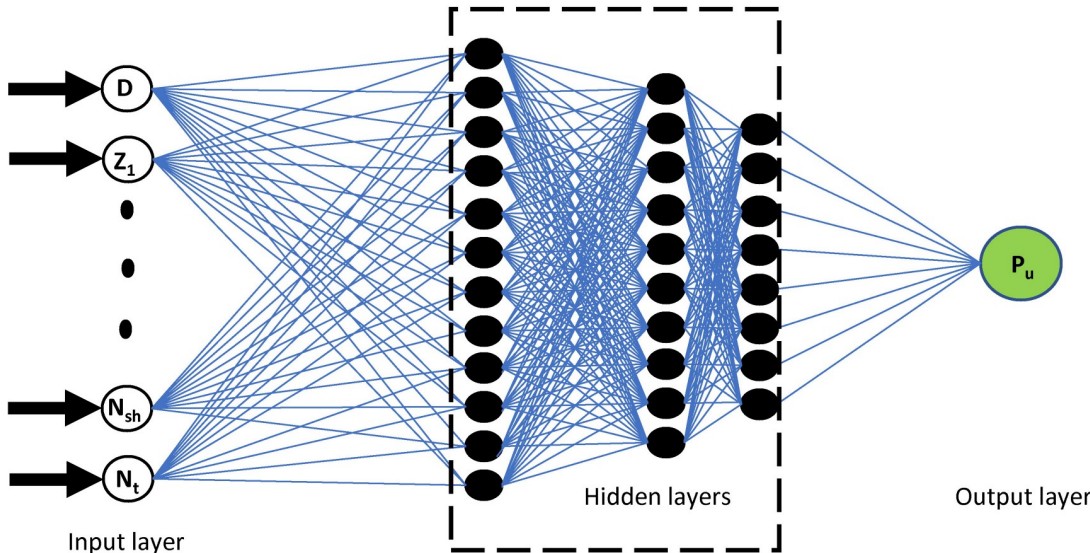

**Fig 4. Illustration of the DLNN used in this study, including 10 inputs, three hidden layers, and one output variable.**

layer perceptron could be called Deep learning neural network (DLNN) [61,62]. In DLNN, each node in a layer is associated with a certain weight, denoted as $w_{ij}$, with every node in the other layers creating a fully linked neural system [63]. Except for the input layer, each node is a neuron that uses a non-linear activation function [64]. Besides, MLP uses a supervised learning technique called backpropagation for the training process [64]. Thanks to its multi-layer, non-linear activation functions, DLNN could distinguish non-linear separable data. Fig 4 shows the DLNN architecture used in this investigation consisting of 10 inputs, three hidden layer and one output variable

A multi-layer perceptron having a linear activation function associated with all neurons represents a linear function network that links the weighted inputs to the output. Using linear algebra, it has been proved that such a network, with any number of layers, can be reduced to a two-layer input-output model. Therefore, the development of the DLNN network using non-linear activation functions is crucial to enhance the accuracy of the model, and better mimic the working mechanism of biological neurons. The use of sigmoid functions is commonly adopted in DLNN network, with two conventional activation functions as below:

$$y(v_i) = \tanh(v_i) \text{ and } y(v_i) = (1 + e^{-v_i})^{-1} \tag{1}$$

The first one represents a hyperbolic tangent, ranges from -1 to 1, whereas the second one is a logistic function with similar shape but ranges from 0 to 1. In these functions, $y(v_i)$ represents the output of the $i^{th}$ node, and $v_i$ is the total weight of the input connection. Besides, alternative activation functions, such as the rectifier, or more specialized function, namely radial basis functions, are also proposed.

In function of the errors of the output compared with the target, the connection weights and biases are adjusted, making the learning process occurs. This could be considered as an example of the supervised learning process using the least-squares average algorithm, which is generalized as a backpropagation algorithm. Precisely, an error in the output node $j$ in the $n^{th}$ data point is given by:

$$e_j(n) = d_j(n) - y_j(n) \tag{2}$$

where $d$ refers to the target value, $y$ denotes the value generated by the perceptron system. The following expression relies on error correction to minimize errors of the predicted output to determine the node weights:

$$\varepsilon(n) = \frac{1}{2} \sum_j e_j^2(n) \tag{3}$$

Furthermore, the following expression uses the gradient descent algorithm to calculate the change, or the correction, for each weight:

$$\Delta\omega_{ji}(n) = -\eta \frac{\partial \varepsilon(n)}{\partial v_j(n)} y_i(n) \tag{4}$$

where $y_i$ denotes the output of the previous neuron, refers to the learning rate. These parameters are chosen to ensure that the error quickly converges without oscillation. Besides, the derivative is calculated based on the local field induced $v_j$, which can be expressed as:

$$-\frac{\partial \varepsilon(n)}{\partial v_j(n)} = e_j(n)\phi'(v_j(n)) \tag{5}$$

where $\phi'$ is the derivative of the activation function. With the change in weight associated with

a hidden node, the relevant derivative can be shown as:

$$-\frac{\partial \varepsilon(n)}{\partial v_j(n)} = \phi'(v_j(n)) \sum_k -\frac{\partial \varepsilon(n)}{\partial v_k(n)} \omega_{kj}(n) \qquad (6)$$

This function depends on the weight changes of nodes representing the $k_{th}$ output layer. This algorithm reflects the inverse backpropagation process, as the output weights change according to the activation function derivative, then the weights of the hidden layer change accordingly.

## 4.2. Genetic Algorithm (GA)

Holland was the first researcher who proposed a genetic algorithm (GA), a stochastic search algorithm, and optimization technique [65]. Later, GA has been investigated by other scientists, especially Deb *et al.* [66], Houck *et al.* [67]. Generally, GA is considered a simple solution for complex non-linear problems [68]. The basis of the method lies in the process of mating, breeding in an initial population, along with several activities such as selection, cross-exchange, and mutation, which help to create new, more optimal individuals [69]. In GA algorithm, the population size is an important factor reflecting the total number of solutions and significantly affects the results of the problem [70], whereas the so-called "generations" refers to the iterations of the optimization process. This process could be conditioned by several selected stopping criteria [71].

Practically, GA method has shown many benefits in finding an optimal resource set to optimize both cost and production [69]. In the field of construction, especially when evaluating the load capacity of piles, many studies have successfully and efficiently used GA method. As an example, Ardalan *et al.* [72] have used GA algorithm combined with neural network to predict driven piles unit shaft resistance from pile loading tests. In another study, 50 PDA (Pile Driving Analyzer) restriction tests were conducted on pre-cast concrete piles to predict the pile bearing capacity. The proposed hybrid method has provided excellent results with $R^2$ of 0.99 [71]. Moreover, other studies on the behavior of piles in soil using the GA method whose effectiveness has been clearly proven [68,70,72–74].

In this work, taking advantage of the GA algorithm, such an optimization technique was used to optimize DLNN to predict the bearing capacity of driven pile. The pseudo algorithm is summarized below (Table 2):

## 4.3. Features selection with GA

It is well-known that the training process with DLNN is a time-consuming and costly method due to the use of computer resource procession [75,76]. In addition, some features in the

**Table 2. Pseudo algorithm of the GA algorithm used in this study.**

```
FOR each chromosome i in Population
  For each gene j
    Initialize Gij randomly within a permissible range
  End FOR
End FOR
Generation k = 1
DO
  FOR each chromosome i in Population
    Calculate the fitness value of Gi
  End FOR
  Mating the best chromosomes to produce more children
  Mutates some children randomly to attempt to find even better candidates
  Remove the weakest chromosomes, based on fitness value, from the Population
k = k + 1
WHILE maximum generation
```

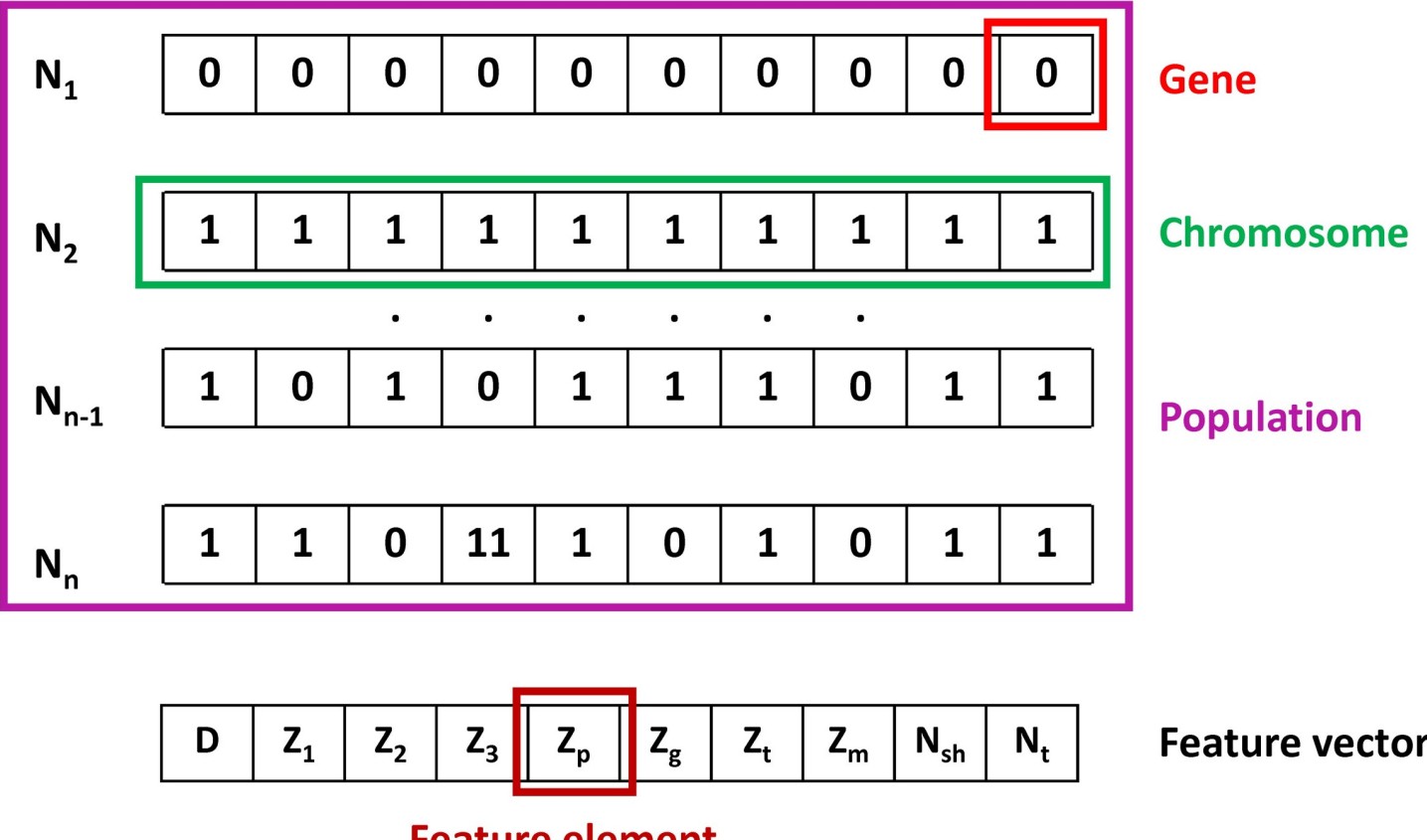

**Fig 5. Representation chromosome of features selection.**

dataset might affect the regression results, as well as unnecessary features might generate noises and reduce prediction accuracy [77]. The selection of appropriate features requires considerable effort, for instance, sum of combinations C(10,i) for i from 1 to 10 could be generated with a dataset containing 10. In order to facilitate the feature selection process, the GA algorithm was used to choose the appropriate features within the dataset, expecting that fewer input variables could enhance the prediction accuracy of GA-DLNN. The detailed process of the selection mechanism is summarized in the following parts.

Firstly, genes inside the chromosome should be selected. In this study, each feature affecting the pile bearing capacity is considered as a gene. As a result, the length of the chromosome is 10, corresponding to 10 features, or 10 genes (Fig 5).

Considering the chromosome, each gene is associated with a unique value, i.e., 1 when it is selected or 0 in the other case [78]. Next, to create the population, original chromosomes are randomly selected [78]. After that, several parents were chosen for mating to create offspring chromosomes based on their fitness value associated with each solution (i.e., chromosome). The fitness value is calculated using a fitness function. The support vector regression (SVR) is chosen as the fitness function for this investigation. In the next step, the regression model is trained with the training dataset, and evaluated on the validation (or testing) dataset. In this study, the mean absolute error (MAE) cost function was used to evaluate the accuracy of the fitness function. The lower the fitness value shows a better solution. Based on the fitness function, the "parents" are filtered from the current population. The nature of GA lies in the

hypothesis that mating two good solutions could produce the best solution [79]. Children born to parents can randomly choose their parents' genes. Mutations are then applied to make new genes in the next generation.

## 4.4. Evolution of DLNN parameters using GA and parameters tuning process

It is universally challenging to find out an optimal neural network architecture. A broad and continuous discussion of this problematic work has been the subject of intense researches. To date, no universal rules are given to define the proper number of hidden layers, neurons in each hidden layer, or functions that connecting the neurons. Considering that in the DLNN algorithm, various possibilities could be assembled to build the final network structure, the selection process becomes unachievable. To overcome this problem, the GA could be used to find the best DLNN architecture in an automatic manner. The mechanism of GA could be summarized as the following.

Firstly, the genes inside the chromosome are determined. Four parameters to be investigated are selected, including (i) the network optimizer algorithm, (ii) the activation function of the hidden layers, (iii) the number of hidden layers, and (iv) the number neurons in each hidden layer. As the number of neurons in each hidden layer is different, more genes are required. Each gene contains data representing the number of neurons in each hidden layer. Considering the maximum number of hidden layers is $P_2$, then the maximum length of the chromosome is $L = (3 + P_2)$. In particular, the first three genes refer to the first three parameters of the model, previously presented. It is worth noticing that in this case, each chromosome has a different length, depending on the corresponding number of hidden layers. Hence, the parameters used for the DLNN architecture could be depicted in Fig 6, such as network optimizer algorithm ($P_0$), the activation function of hidden layers ($P_1$), the number of hidden layers ($P_2$), and the number neurons in each hidden layer ($P_3$...$P_L$).

The considered fitness function is DLNN model, along with four cost functions to evaluate the performance, namely $R^2$, IA, MAE, and RMSE. Detailed descriptions of these criteria are given in the next section. Given that the length of the chromosome might be different, the mating progress occurs under the following principles:

(i). If the length of the parents' chromosomes is similar, the child will randomly select the number of hidden layers and the number of neurons from father or mother.

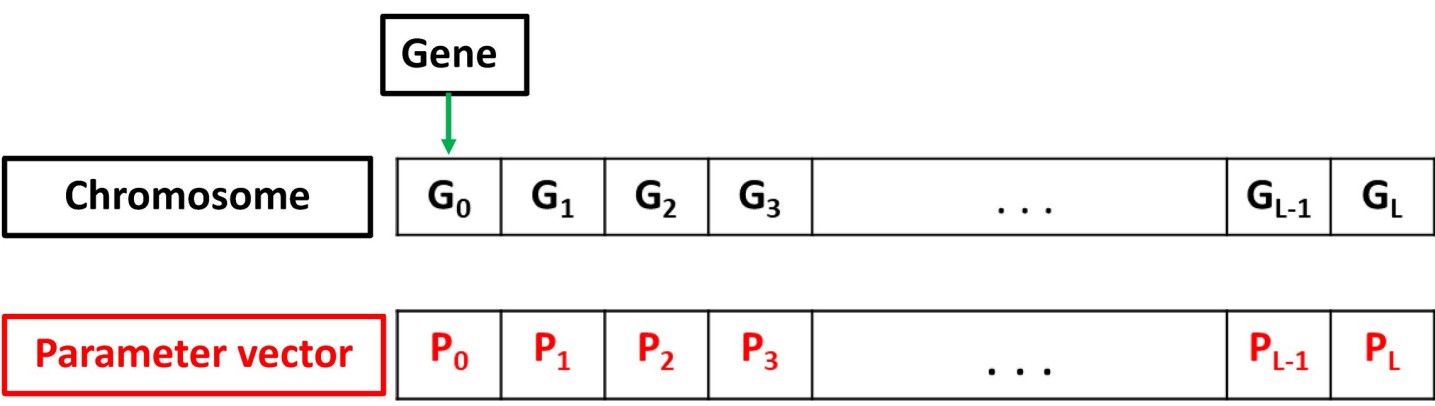

**Fig 6. Chromosome representation of the parameters selection process.**

(ii). If the length of the parents' chromosomes is different, two cases could be considered in this case. In the first case, supposing the child chooses the number of hidden layers from a person with fewer genes, the selection will be random from the parents. In the second case, a child chooses to take the number of hidden layers from a parent that has more genes. The only option is to select the missing gene from a person with a higher chromosome length, and other genes are taken randomly from their parents. The mating process is highlighted in Fig 7.

During the mutation process, few children are selected. Besides, a random gene is selected and replaced with another random value within a given range. Particularly, since the DLNN model has many parameters, the mutation rate is set at 50% of the number of children born in order to maximize the chance to find the best genes. Finally, the parameters of DLNN were finely tuned by GA through population generations to find out the best prediction performance. Table 3 summarizes the tuned parameters and their tuning ranges and options.

## 4.5. Performance evaluation

In order to verify the effectiveness and performance of the ML algorithms, four different criteria were selected in this study, namely, root mean square error (RMSE), mean absolute error (MAE), the coefficient of determination ($R^2$), and Willmott's index of agreement (IA). The criterion RMSE is the mean squared difference between the predicted outputs and targets, whereas MAE is the mean magnitude of the errors. The similarity between the two error criteria RMSE and MAE is that the closer these errors' criterion values to 0, the better performance of the model. The criterion $R^2$ is the correlation between targets and outputs [80]. The accuracy of the model is superior in the cases of small values of RMSE and MAE. The values of $R^2$ are in the range of $[-\infty \div 1]$, where higher accuracy is obtained when the values are close to 1. The Index of Agreement (IA) was presented by Willmott [81,82]. The IA points out the ratio of the mean square error and the potential error. Similar to $R^2$, the values of IA vary between $-\infty$ and 1, in which 1 indicates a perfect correlation, and negative value indicates no agreement. These coefficients can be calculated using the following

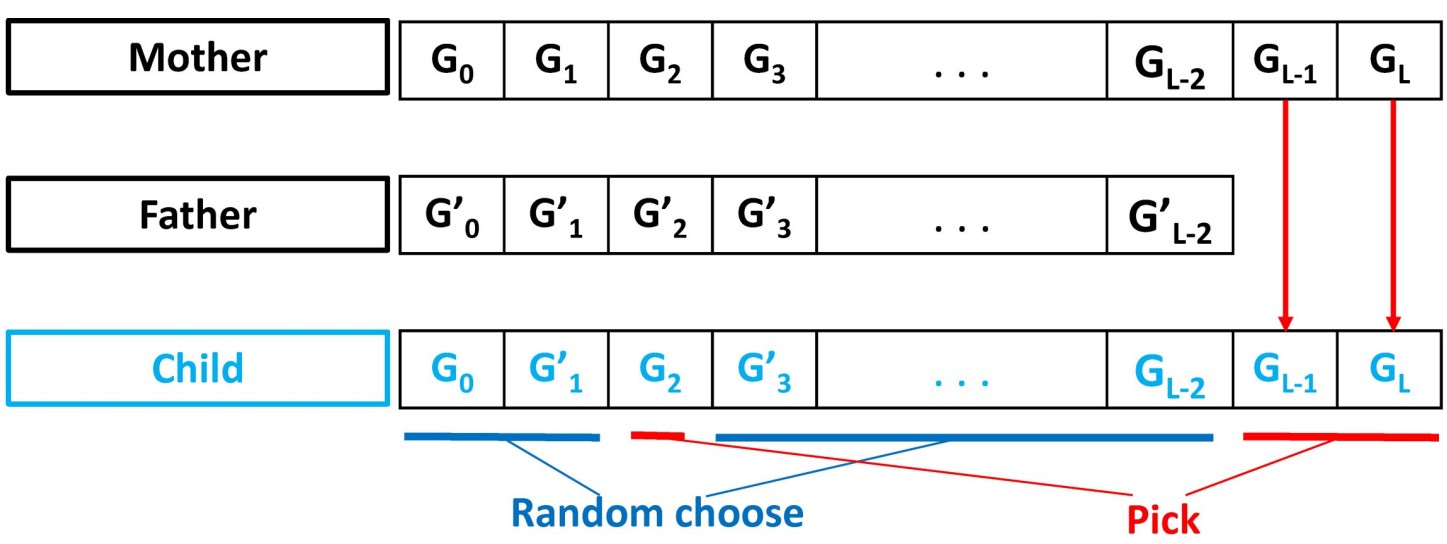

**Fig 7. The mating process with different chromosomes length.**

Table 3. Parameters of DLNN and their tuning ranges/options to be optimized by GA.

| N° | Parameter | Explanation | Range/Option |
|---|---|---|---|
| 1 | $P_0$ | Network optimizer algorithm | Quasi-Newton, Stochastic gradient descent, Adam |
| 2 | $P_1$ | Activation function of hidden layers | Identity, Logistic, Tanh, Relu |
| 3 | $P_2$ | Number of hidden layers | 2–10 |
| 4 | $P_3$ | Number neurons in hidden layer 1 | 2–80 |
| 5 | $P_4$ | Number neurons in hidden layer 2 | 2–80 |
| . | . | . | . |
| . | . | . | . |
| L—1 | $P_{L-1}$ | Number neurons in hidden layer (L-3) | 2–80 |
| L | $P_L$ | Number neurons in hidden layer (L-2) | 2–80 |

L = Length of the chromosome, L = (3 + $P_2$).

formulas [83,84]:

$$MAE = \frac{1}{k}\sum_{i=1}^{k}(v_i - \bar{v}_i) \tag{7}$$

$$RMSE = \sqrt{\frac{1}{k}\sum_{i=1}^{k}(v_i - \bar{v}_i)} \tag{8}$$

$$R^2 = 1 - \frac{\sum_{i=1}^{k}\left(v_i - \bar{v}_i\right)^2}{\sum_{i=1}^{k}\left(v_i - \bar{v}\right)^2} \tag{9}$$

$$IA = 1 - \frac{\sum_{i=1}^{N}\left(v - \bar{v}_i\right)^2}{\sum_{i=1}^{N}\left(|v - \bar{v}| + |v_i - \bar{v}|\right)^2} \tag{10}$$

Where k inferred the number of the samples, $v_i$ and $\bar{v}_i$ were the actual and predicted outputs, respectively, and $\bar{v}$ was the average value of the $v_i$.

## 5. Results and discussion

### 5.1. Feature selection

The results of the feature selection process using the GA model is presented in this section. The initialization parameters of GA used in this study are given in Table 4. Fig 8 illustrates the evolution of MAE values using GA after 200 generations. It can be seen that the MAE value was progressively decreased with the generation of GA. The lowest MAE was 116.91 (kN) at the first generation and decreased to 95.54 (kN) at the 87th generation. This value was unchanged from the 87th to the 200th generation. The optimum representation chromosome of feature selection were [0, 1, 1, 0, 0, 1, 0, 0, 0, 1]. This result suggested a new dataset, more compact, corresponded to [$Z_1$, $Z_2$, $Z_g$, $N_t$]. Therefore, the number of input variables for a

**Table 4. GA feature selection initialization parameters.**

| Parameters | Value and Description |
|---|---|
| Number of population | 25 |
| Number of generation | 200 |
| Mating pool size | 10 |
| Mutation rate | 0.5 |
| Fitness function | Support Vector Regression (SVR) |
| Cost function | MAE |
| Data used | Training/ Validation dataset |

compact dataset included 4 variables. As a result, the input space was reduced by 6 variables compared to the original dataset.

## 5.2. Optimization of DLNN architecture

The evolutionary results in predicting the pile bearing capacity of GA-DLNN model are evaluated in this section. The initialization parameters of GA-DLNN used in this study are given in Table 5. Fig 9 illustrates the evolution of the GA-DLNN model through 200 generations with 4 and 10 input variables. A summary of the best predictability of the models is presented in Table 6. For the sake of conparison and highlight the performance of the reduced input space, three different scenarios were performed. The first one used the 4-input space and simulated with GA-DLNN, denoted as 4-input GA-DLNN model. The second one contained the initial input space and performed with GA-DLNN, denoted as the 10-input GA-DLNN model. The last scenario referred to the case using 4 input variables but using DLNN as a predictor, denoted as 4-input DLNN model.

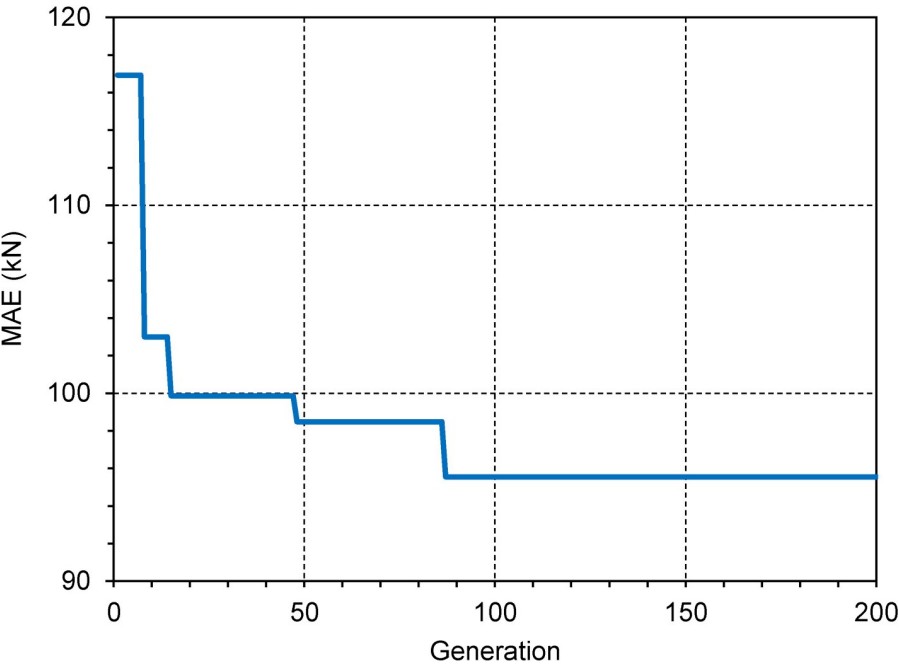

**Fig 8. Features selection using the GA model.**

**Table 5. GA-DLNN initialization parameters.**

| Parameters | Value and Description |
|---|---|
| Number of population | 25 |
| Number of generation | 200 |
| Mating pool size | 24 |
| Mutation rate | 0.5 |
| Fitness function | DLNN |
| Cost function | $R^2$, MAE, RMSE, IA |
| Data used | Training/ Validation dataset |

It can be seen that the 4-input GA-DLNN model performed better accuracy, the best generation yielded correlation of $R^2$ = 0.923, MAE = 75.927, RMSE = 95.118 and IA = 0.981. Compared to the first generation, the 4-input DLNN model produce accurate intermediate precision ($R^2$ = 0.858, MAE = 90.785, RMSE = 123.788 and IA = 0.967).

The results also show that the 4-input GA-DLNN model gives slightly better performance than the 10-input GA-DLNN model. The GA-DLNN model with 10 variables predicts

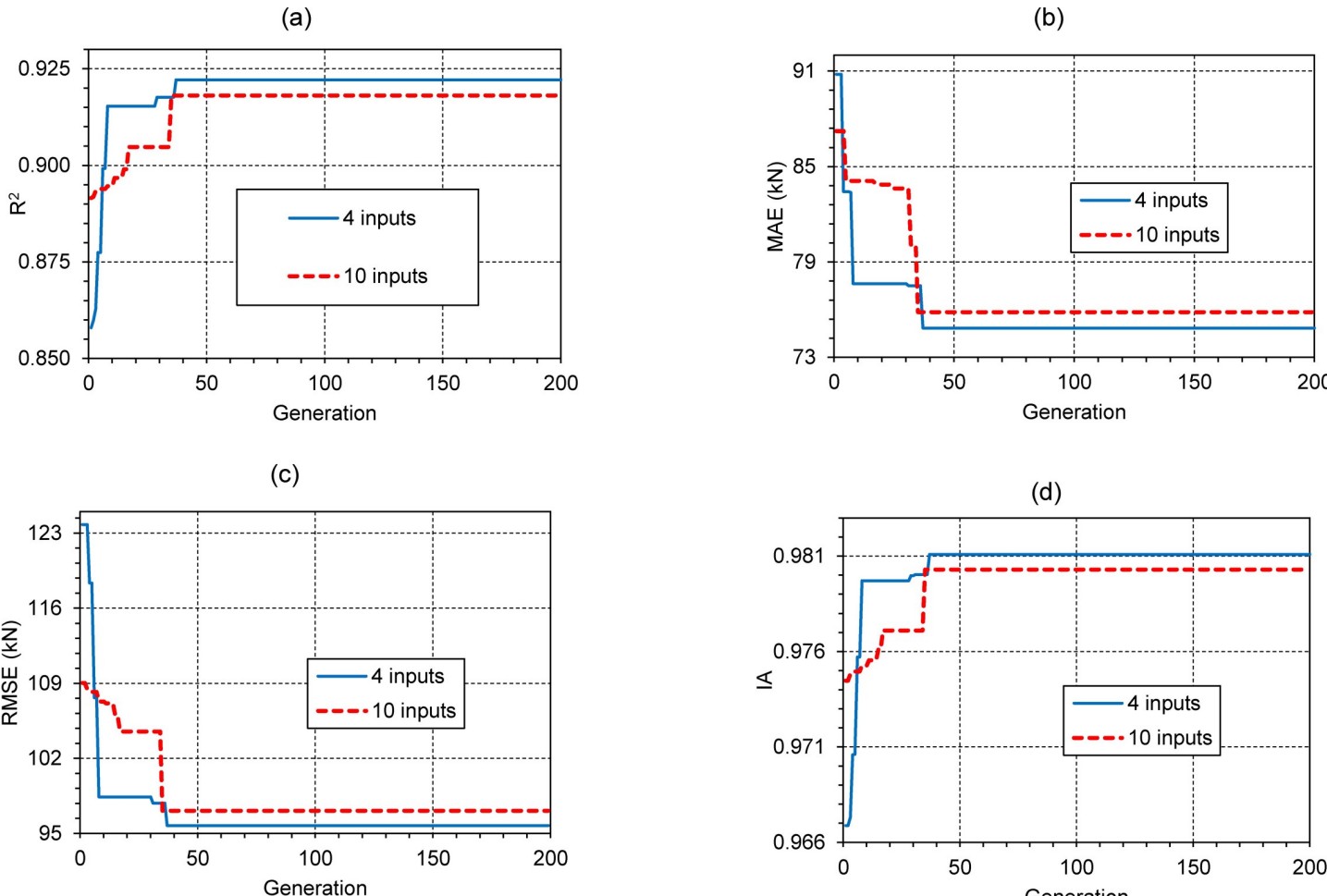

**Fig 9. Parameters tuning using the model using the GA-DLNN model with 4 and 10 inputs.**

**Table 6. Summary of best prediction capability of models.**

| Model | $R^2$ | MAE (kN) | RMSE (kN) | IA | Normalized time |
|---|---|---|---|---|---|
| 4-input GA-DLNN | 0.923 | 75.927 | 95.118 | 0.981 | 0.7 |
| 10-input GA-DLNN | 0.918 | 75.838 | 97.092 | 0.980 | 1.0 |
| 4-input DLNN | 0.858 | 90.785 | 123.788 | 0.967 | - |

correlation results at most efficient generation as follows: R2 = 0.918, MAE = 75.838, RMSE = 97.092 and IA = 0.980. The analysis time cost through 200 generations of the 4-input model is much lower than the 10-input model with the normalized time of the two models, respectively: 0.7 and 1.0.

The optimum parameters of models are shown in Table 7. It shows that all three models choose the same network optimization algorithm (Quasi-Newton), the number of hidden layers range from 2 to 4 and the number of neurons in each hidden layer is relatively complex, ranging from 9 to 80. However, each model chooses a different type of activation function.

## 5.3. Predictive capability of the models

Fig 10 shows a visual comparison of test results and predictions based on Pu from a representative ML model. The performance of ML models has been tested on all three datasets: training, validation and testing. In this case, two representative DLNN models were selected based on the best performance through the model evolution (Fig 9), corresponding to input variables 4 and 10. One 4-input DLNN model which has the best fitness value in the first generation, was chosen to compare with the two optimal models to prove the effectiveness of model evolution. The predictive capability of the models is also summarized in Table 8.

From a statistical standpoint, the performance of ML algorithms should be fully evaluated. As mentioned during the simulation, 60% of the test data was randomly selected to train ML models. The performance of such a model can be affected by the selection order of the training data set. Therefore, a total of 1000 simulations were performed next, taking into account the random splitting effect in the dataset. The result is shown in Fig 11 and Tables 9–12. It can be seen that the performance of the 4-input GA-DLNN model was improved after tuning the parameters of the DLNN model and outperformed the best model in the first generation (4-input DLNN). On training set, R2 value has increased from 0.919 to 0.932. The result can be also observed on the validation set, in which the R2 value is increased (from 0.884 to 0.898). The most difference can be seen in the testing set in which R2 increased from 0.777 to 0.882. Compared to the 10-input GA-DLNN model, the R2 value is similar in training and validation, the big difference only appears in the test data set, whereas R2 value of the 4-input GA-DLNN model gives better results (R2 = 0.882) compared to 10-input GA-DLNN models (R2 = 0.8). On testing set, SD value of 4-input GA-DLNN model is smallest (SD = 0.008) compare to 10-input GA-DLNN and 4-input DLNN model (SD = 0.0351, 0.0718, respectively), indicating more stable 4-input GA-DLNN modelling.

**Table 7. The optimum parameter of models.**

| Parameter | 4-input GA-DLNN | 10-input GA-DLNN | 4-input DLNN |
|---|---|---|---|
| Network optimizer algorithm | Quasi-Newton | Quasi-Newton | Quasi-Newton |
| Activation function of hidden layers | logistic | relu | relu |
| Number of hidden layers | 2 | 4 | 3 |
| Number neurons in each hidden layer | (33, 80) | (74, 17, 24, 12) | (9, 50, 29) |

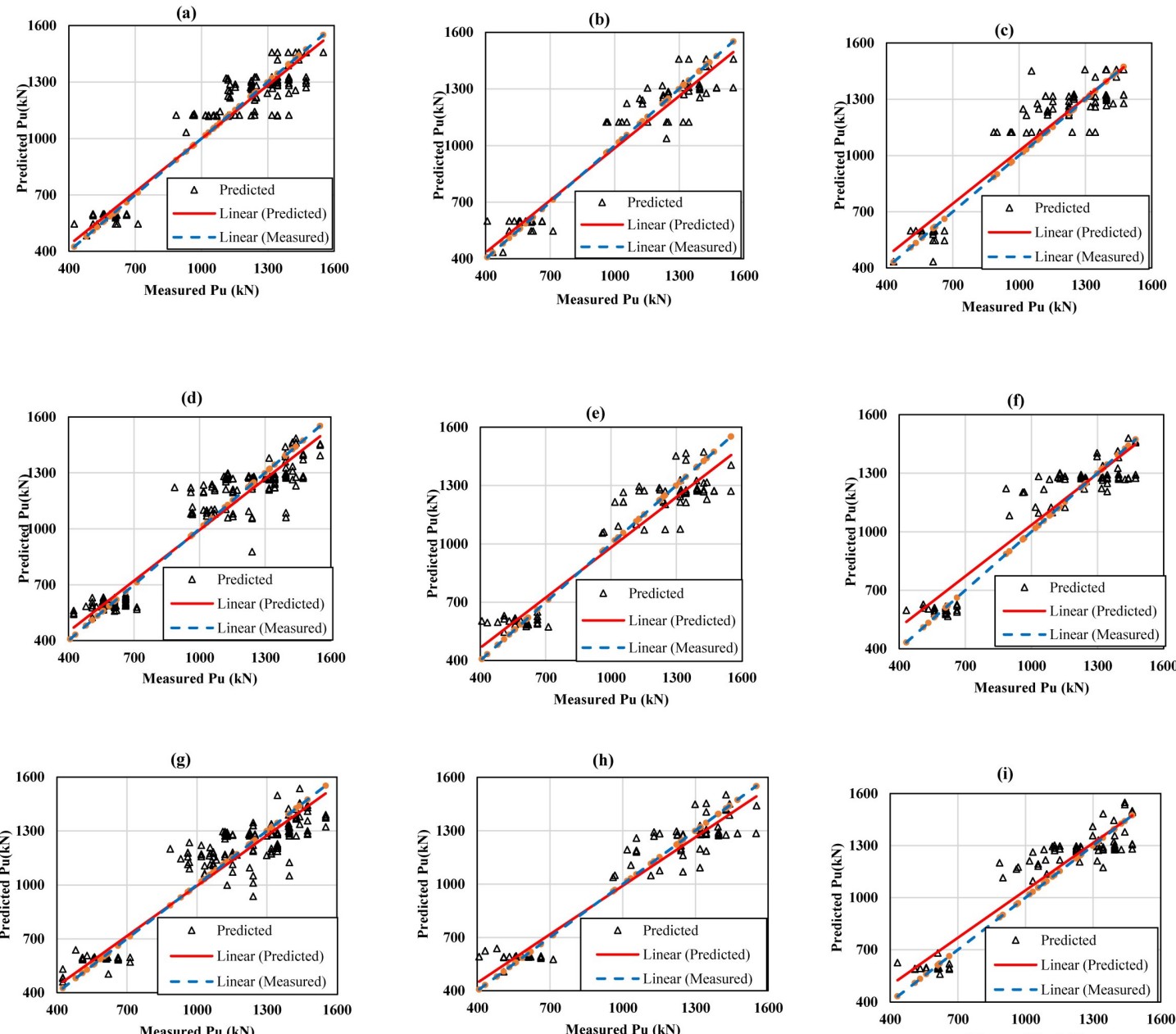

**Fig 10.** Measured and predicted values of axial bearing capacity of pile using the models: 4-input GA-DLNN model for training (a), validation (b), testing dataset (c); 10-input GA-DLNN model for training (d), validation (e), testing dataset (f); 4-input DLNN model for training (g), validation (h), testing dataset (i).

Table 13 presents some research results on ML applications in foundation engineering. The results of this study as well as previous studies show that the expected foundation effectiveness of ML technique in foundation engineering with prediction results of foundation load is mostly reaching R2 from 0.8 to 0.9 on test data set. However, due to the use of different data sets, a comparison between these results is unwarranted. A project that uses different data sets is needed to give a generalized model to foundation engineering.

**Table 8. Predictive capability of the models.**

| Dataset | Cost function | 4-input GA-DLNN | 10-input GA-DLNN | 4-input DLNN |
|---------|---------------|-----------------|------------------|--------------|
| Training | $R^2$ | 0.944 | 0.927 | 0.910 |
| | MAE | 64.235 | 72.744 | 75.929 |
| | RMSE | 83.593 | 94.873 | 105.884 |
| | IA | 0.985 | 0.981 | 0.976 |
| Validation | $R^2$ | 0.923 | 0.918 | 0.858 |
| | MAE | 75.927 | 75.838 | 90.785 |
| | RMSE | 95.118 | 97.092 | 123.788 |
| | IA | 0.981 | 0.980 | 0.967 |
| Testing | $R^2$ | 0.887 | 0.844 | 0.809 |
| | MAE | 86.573 | 93.074 | 92.867 |
| | RMSE | 110.176 | 132.490 | 142.896 |
| | IA | 0.969 | 0.956 | 0.947 |

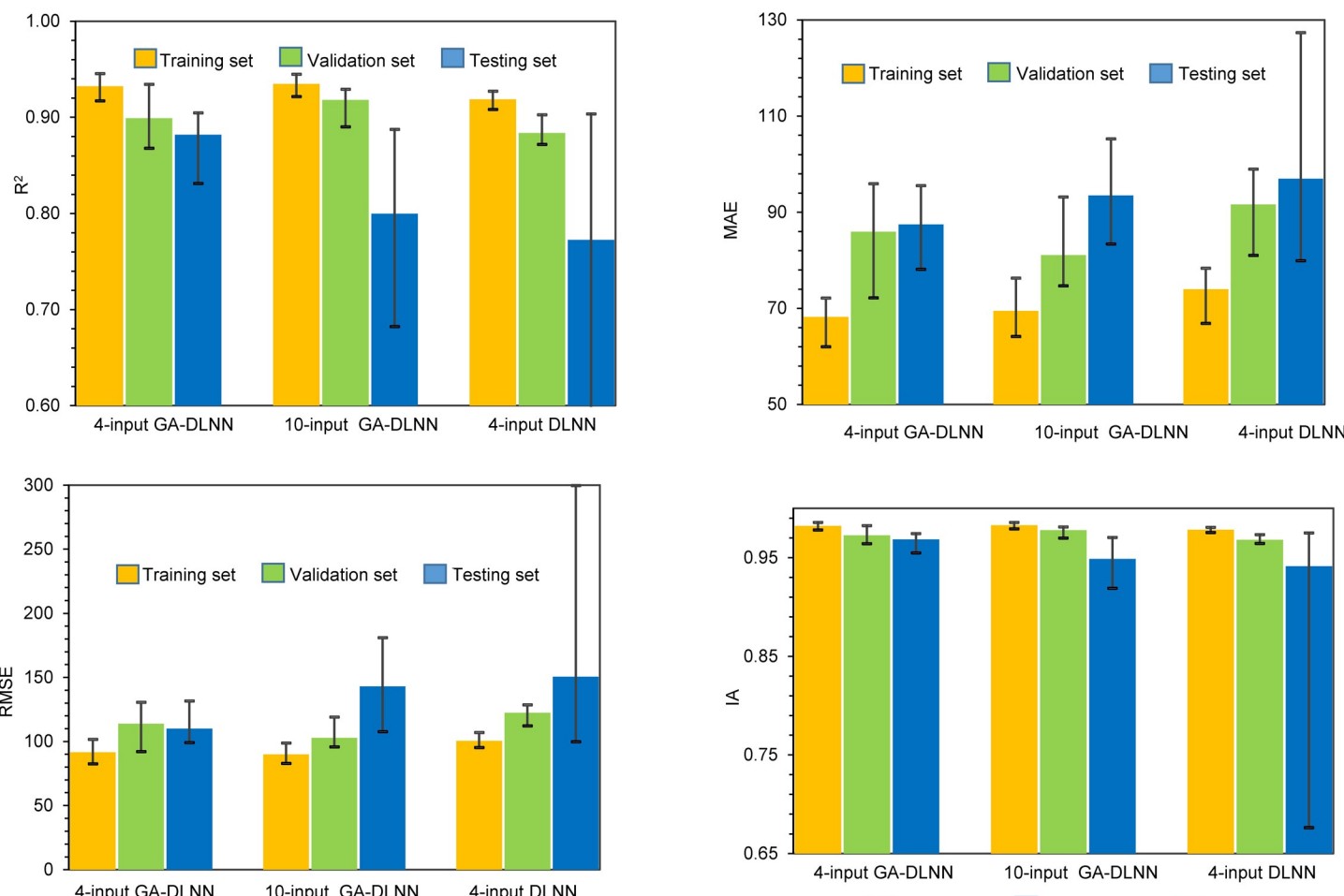

**Fig 11. Predictive capability of the models with 1000 simulations.**

**Table 9. Summary of the 1000 simulations using $R^2$ criteria.**

| Model | Dataset | Average | Min | Max | SD |
|---|---|---|---|---|---|
| 4-input GA-DLNN | Training | 0.932 | 0.917 | 0.945 | 0.0073 |
| | Validation | 0.899 | 0.868 | 0.934 | 0.0138 |
| | Testing | 0.882 | 0.831 | 0.905 | 0.0082 |
| 4-input DLNN | Training | 0.919 | 0.905 | 0.928 | 0.0038 |
| | Validation | 0.884 | 0.872 | 0.905 | 0.0040 |
| | Testing | 0.777 | 0.514 | 0.902 | 0.0718 |
| 10-input GA-DLNN | Training | 0.924 | 0.907 | 0.932 | 0.0052 |
| | Validation | 0.918 | 0.895 | 0.930 | 0.0054 |
| | Testing | 0.800 | 0.671 | 0.890 | 0.0351 |

**Table 10. Summary of the 1000 simulations using IA criteria.**

| Model | Dataset | Average | Min | Max | SD |
|---|---|---|---|---|---|
| 4-input GA-DLNN | Training | 0.982 | 0.978 | 0.986 | 0.0020 |
| | Validation | 0.973 | 0.964 | 0.982 | 0.0038 |
| | Testing | 0.969 | 0.955 | 0.974 | 0.0021 |
| 4-input DLNN | Training | 0.978 | 0.975 | 0.981 | 0.0010 |
| | Validation | 0.968 | 0.964 | 0.973 | 0.0012 |
| | Testing | 0.941 | 0.676 | 0.975 | 0.0215 |
| 10-input GA-DLNN | Training | 0.983 | 0.979 | 0.986 | 0.0009 |
| | Validation | 0.978 | 0.970 | 0.981 | 0.0017 |
| | Testing | 0.949 | 0.919 | 0.970 | 0.0088 |

**Table 11. Summary of the 1000 simulations using RMSE criteria.**

| Model | Dataset | Average | Min | Max | SD |
|---|---|---|---|---|---|
| 4-input GA-DLNN | Training | 91.537 | 82.268 | 101.353 | 4.9118 |
| | Validation | 113.764 | 91.962 | 130.376 | 7.8739 |
| | Testing | 109.965 | 98.902 | 131.414 | 3.7948 |
| 4-input DLNN | Training | 100.444 | 95.014 | 106.750 | 2.3025 |
| | Validation | 122.405 | 112.008 | 128.483 | 2.0521 |
| | Testing | 150.528 | 99.600 | 299.543 | 27.3845 |
| 10-input GA-DLNN | Training | 89.901 | 82.692 | 98.562 | 2.3062 |
| | Validation | 102.720 | 95.497 | 118.908 | 3.4459 |
| | Testing | 143.002 | 107.493 | 180.817 | 12.7996 |

**Table 12. Summary of the 1000 simulations using MAE criteria.**

| Model | Dataset | Average | Min | Max | SD |
|---|---|---|---|---|---|
| 4-input GA-DLNN | Training | 68.211 | 61.977 | 72.091 | 1.7106 |
| | Validation | 85.937 | 72.163 | 95.921 | 3.3777 |
| | Testing | 87.459 | 78.075 | 95.510 | 2.2845 |
| 4-input DLNN | Training | 73.960 | 66.832 | 78.319 | 2.2208 |
| | Validation | 91.629 | 80.999 | 98.967 | 2.9166 |
| | Testing | 96.997 | 79.877 | 127.320 | 5.6667 |
| 10-input GA-DLNN | Training | 69.458 | 64.106 | 76.256 | 1.6571 |
| | Validation | 81.074 | 74.631 | 93.133 | 2.8590 |
| | Testing | 93.507 | 83.376 | 105.197 | 3.1047 |

**Table 13. Comparison with other studies.**

| Author | Model | Foundation type | Number of samples | $R^2$ | RMSE |
|---|---|---|---|---|---|
| Momeni el al. [56] | ANFIS | Thin-walls | 150 | 0.875 | 0.048 |
| | ANN | | | 0.71 | 0.529 |
| Momeni el al. [85] | GPR | Piles | 296 | 0.84 | - |
| Kulkarni el al. [86] | GA-ANN | Rock-socketed piles | 132 | 0.86 | 0.0093 |
| Jahed Armaghani el al. [87] | ANN | | | 0.808 | 0.135 |
| | PSO-ANN | | | 0.918 | 0.063 |
| The present study | GA-DNN | Piles | 472 | 0.882 | 109.965 |

## 6. Conclusions

The main achievement of this study is to provide an efficient GA-DLNN hybrid model in predicting pile load capacity. The model has the ability to self-evolve to find the optimal model structure, where the optimal number of hidden layers can be treated as a variable and discovered during the model's evolution, besides to the other important parameters. In addition, an evolutionary model was developed to mitigate the number of input variables of the model, while ensuring the accuracy of the regression results.

The results showed that, on the training data set, all three models: 4 -input GA-DLNN, 10-input GA-DLNN and 4-input DLNN have good predict results, in which, the leading is the model GA-DLNN with 4 inputs. On the validation data set, the 4-input GA-DLNN model gave similar results to the 10-input GA-DLNN model and outperformed the 4-input DLNN model with satisfactory accuracy (R2 = 0.923, MAE = 75.927, RMSE = 95.118 kN, IA = 0.981 using 4-input GA-DLNN compared with R2 = 0.918, MAE = 75.838 kN, RMSE = 97.092 kN, IA = 0.98 using 10-input GA-DLNN and R2 = 0.858, MAE = 90.785, RMSE = 113.788 kN, IA = 0.967 kN using 4-input DLNN). Meanwhile, the time cost for the 4-input GA-DLNN model is much lower than the 10-input GA-DLNN hybrid model (the normalize time is respectively 0.7 and 1.0). On testing data, the predictability of the 4-input GA-DLNN model proved to be superior to the other two models. The forecast result of 1000 simulations shows that the average value of R2 is 0.882, 0.8, 0.777 respectively for 4-input GA-DLNN models, 10-input GA-DLNN and 4-input DLNN. In addition, the oscillation range (minimum, maximum) of R2 value of input model GA-DLNN 4 is smaller than the other 2 models, indicating the model's stability.

As research shows that the best results are obtained by GA-DLNN with the number of hidden layers from 2 to 4. The number of neurons in each hidden layer is completely different and is distributed complexly in the hidden layers. It suggests that a DLNN model with 2, 3, 4 hidden layers might be optimal for the problem related to predicting the bearing capacity of driven piles. However, it is recommended to select the number of neurons in each hidden layer by evolutionary methods to bring out high performance for the DLNN model. The results obtained from the evolution of the DLNN model by GA show that the activation function of hidden layers mainly choose one of two categories: relu or logistic and the Quasi-Newton optimal algorithm is most suitable for predicting bearing capacity of pile.

## Supporting information

**S1 File.**
(CSV)

**S1 Appendix.**
(DOCX)

## Author Contributions

**Conceptualization:** Tuan Anh Pham.

**Formal analysis:** Tuan Anh Pham.

**Investigation:** Tuan Anh Pham.

**Validation:** Tuan Anh Pham.

**Writing – original draft:** Tuan Anh Pham, Van Quan Tran, Huong-Lan Thi Vu.

**Writing – review & editing:** Tuan Anh Pham, Van Quan Tran, Hai-Bang Ly.

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
