## [Decision Letter · Decision Letter 0]

6 Oct 2020

PONE-D-20-26359

Design Deep Neural Network Architecture using a Genetic Algorithm for Estimation of Pile Bearing Capacity

PLOS ONE

Dear Dr. Tran,

Thank you for submitting your manuscript to PLOS ONE. After careful consideration, we feel that it has merit but does not fully meet PLOS ONE’s publication criteria as it currently stands. Therefore, we invite you to submit a revised version of the manuscript that addresses the points raised during the review process.

We look forward to receiving your revised manuscript.

Kind regards,

**Le Hoang Son, Ph.D**

Academic Editor

PLOS ONE

2. In your Data Availability statement, it is unclear why you have selected the option 'No - some restrictions will apply'. We note that you have indicated that data from this study are available upon request. PLOS only allows data to be available upon request if there are legal or ethical restrictions on sharing data publicly. For more information on unacceptable data access restrictions, please see http://journals.plos.org/plosone/s/data-availability#loc-unacceptable-data-access-restrictions.

4.  We note that Figure 1 in your submission contain map images which may be copyrighted. All PLOS content is published under the Creative Commons Attribution License (CC BY 4.0), which means that the manuscript, images, and Supporting Information files will be freely available online, and any third party is permitted to access, download, copy, distribute, and use these materials in any way, even commercially, with proper attribution. For these reasons, we cannot publish previously copyrighted maps or satellite images created using proprietary data, such as Google software (Google Maps, Street View, and Earth). For more information, see our copyright guidelines: http://journals.plos.org/plosone/s/licenses-and-copyright.

4.1.    You may seek permission from the original copyright holder of Figure 1 to publish the content specifically under the CC BY 4.0 license. 

4.2.    If you are unable to obtain permission from the original copyright holder to publish these figures under the CC BY 4.0 license or if the copyright holder’s requirements are incompatible with the CC BY 4.0 license, please either i) remove the figure or ii) supply a replacement figure that complies with the CC BY 4.0 license. Please check copyright information on all replacement figures and update the figure caption with source information. If applicable, please specify in the figure caption text when a figure is similar but not identical to the original image and is therefore for illustrative purposes only.

5. We note you have included a table to which you do not refer in the text of your manuscript. Please ensure that you refer to Table 9, 10, 11 in your text; if accepted, production will need this reference to link the reader to the Table.

**Comments to the Author**

1. Is the manuscript technically sound, and do the data support the conclusions?

Reviewer #1: Yes

Reviewer #2: Yes

Reviewer #3: Yes

2. Has the statistical analysis been performed appropriately and rigorously? 

Reviewer #1: Yes

Reviewer #2: Yes

Reviewer #3: Yes

3. Have the authors made all data underlying the findings in their manuscript fully available?

Reviewer #1: No

Reviewer #2: No

Reviewer #3: Yes

4. Is the manuscript presented in an intelligible fashion and written in standard English?

Reviewer #1: Yes

Reviewer #2: Yes

Reviewer #3: Yes

5. Review Comments to the Author

**Reviewer #1**: 

I have read the paper entitled "Design Deep Neural Network Architecture using a Genetic Algorithm for Estimation of Pile Bearing Capacity". In essence, the paper suggests a deep ANN-based predictive model for pile bearing capacity. It is interesting that authors used GA for reducing the number of features from 10 to 4. The paper is well written and well organized. Although compared to the previous publications, slight contribution was observed, presenting new sets of real data is always of interest as it can constitute common sense. Hence, firstly authors are requested to present at least 100 sets of data in the appendix. Further comments are presented in the following lines:

2. Include VAF performance index.

3. Enhance the literature review considerably by providing a Table of previous AI-based works in the field of foundation engineering including deep foundation, shallow foundation, thin-walled foundations below are some recommendations however authors do not have to cite them necessarily if they find them irrelevant. the implemented soft computing technique, type of foundations, dataset size, R or R2 should be highlighted in this table.

4. It should be clearly highlighted in the introduction that in what aspect the presented paper is different from other studies (like implementation of deep learning)

5. despite AI advantages, limitations of these methods should be clearly highlighted.

6. A competitor like conventional BP-ANN is needed for comparison purposes or the prediction performance of the proposed AI-based predictive model should be checked against other works.

7. checking the English is suggested.

Marto, A., Hajihassani, M., & Momeni, E. (2014). Bearing Capacity of Shallow Foundation's Prediction through Hybrid Artificial Neural Networks. In Applied Mechanics and Materials (Vol. 567, pp. 681-686). Trans Tech Publications Ltd.

Momeni, E., Armaghani, D. J., Fatemi, S. A., & Nazir, R. (2018). Prediction of bearing capacity of thin-walled foundation: a simulation approach. Engineering with Computers, 34(2), 319-327.

Momeni, E., Dowlatshahi, M. B., Omidinasab, F., Maizir, H., & Armaghani, D. J. (2020). Gaussian Process Regression Technique to Estimate the Pile Bearing Capacity. Arabian Journal for Science and Engineering, 1-13.

Nazir, R., Momeni, E., Marsono, K., & Maizir, H. (2015). An artificial neural network approach for prediction of bearing capacity of spread foundations in sand. Jurnal Teknologi, 72(3).

Rezaei, H., Nazir, R., & Momeni, E. (2016). Bearing capacity of thin-walled shallow foundations: an experimental and artificial intelligence-based study. Journal of Zhejiang University-SCIENCE A, 17(4), 273-285.

**Reviewer #2**: 

Introduction: As there are plenty of studies involving the GA optimized DNNs in this field I strongly advise to explain the novelty clearly and justify the need for this particular research.

Section 2.2. Data preparation: line 2 "[...] all the factors affecting the pile bearing capacity were considered.". I suggest to put it that way "all the known factors" as all the factors affecting the bearing capacity might not be discovered yet.

Section 4.2. Optimization of DLNN Architecture: line 10 "[...] model performed well better performance"?

Section 4.3. Predictive Capability of the Models: In Tab. 7. you compare the "predictive capability of the models" on three datasets (training, validation and testing). Low error achieved on the training and validation dataset does not mean that the model will predict accurately (i.e. perform good on testing set). When the function fits the training data very well the model's predictions can often be not so accurate (overfitting), cause the model has lower generalization ability. Therefore the predictive capability of the model can only be measured with the error obtained on the testing dataset.

Conclusions: I suggest pointing out the main achievement of this study, maybe mentioning possible applications of the developed model and future research perspectives.

**Reviewer #3**: 

The manuscript describes a technically sound scientific study with data that supports the conclusions. The experiments were performed rigorously with appropriate controls (four control conditions: R2, IA, RMSE and MAE), replication and sample size (1000 replicates, 3 structure configurations). On the basis of the obtained data, appropriate conclusions were drawn. The statistical analysis was performed appropriately and rigorously, although the presentation of the results shows a lack of consistency in the data:

- Figure 11 shows the R values whose R2 equivalents do not match the values summarized in Table 7.

- The results of the analyzes presented in Tables 9, 10 and 11 are identical and should contain collected values according to different three criteria.

The authors provided graphical access to all the data underlying the findings in their manuscript. The manuscript is clearly presented and written in standard English, but there are some minor typing errors in the text, e.g.:

- page 12: “The initialization parameters of GA used in this study are given in Tables 3.” (should be: “… in Table 3.”).

- page 21: “On the validation data set, the 4-input GA-DLNN model gave similar results to the 10-input GA-DLNN model and outperformed the 4-input GA-DLNN model with satisfactory accuracy…” (should be: “… the 4-input DLNN model with satisfactory accuracy…”).

The authors of the manuscript made every effort to ensure that the final version of the text was of the highest possible scientific and editorial level, but the reviewer believes that the combination of graphs presented in Figures 9 and 10 would allow for an easier comparative analysis of the results. The aim of the graphical presentation of the results in Figure 12 was a comparative analysis - according to the reviewer, changing the vertical scale (starting from higher values) will allow to emphasize the differences between the examined structures and the types of tests performed.

---

## [Author Response · Author response to Decision Letter 0]

8 Nov 2020

RESPONSES OF THE ACADEMIC EDITOR AND REVIEWER’S COMMENTS

I. Responses to academic editor

Comment 1: Please ensure that your manuscript meets PLOS ONE's style requirements, including those for file naming. The PLOS ONE style templates can be found at

Response:

We thank the academic editor for your announcements. We have already corrected our manuscript follow styling requests and resubmitted them with the revise version.

Comment 2: In your Data Availability statement, it is unclear why you have selected the option 'No - some restrictions will apply'. We note that you have indicated that data from this study are available upon request. PLOS only allows data to be available upon request if there are legal or ethical restrictions on sharing data publicly. For more information on unacceptable data access restrictions, please see http://journals.plos.org/plosone/s/data-availability#loc-unacceptable-data-access-restrictions.

 Response:

We thank the academic editor for your announcements. We have uploaded the required data which are found in the Appendix section of paper.

Comment 3: Thank you for stating the following financial disclosure:

a. Please clarify the sources of funding (financial or material support) for your study. List the grants or organizations that supported your study, including funding received from your institution.

d. If you did not receive any funding for this study, please state: “The authors received no specific funding for this work.”

 Response:

We added the statement “The authors received no specific funding for this work.” in the cover letter

Comment 4: We note that Figure 1 in your submission contain map images which may be copyrighted. All PLOS content is published under the Creative Commons Attribution License (CC BY 4.0), which means that the manuscript, images, and Supporting Information files will be freely available online, and any third party is permitted to access, download, copy, distribute, and use these materials in any way, even commercially, with proper attribution. For these reasons, we cannot publish previously copyrighted maps or satellite images created using proprietary data, such as Google software (Google Maps, Street View, and Earth). For more information, see our copyright guidelines: http://journals.plos.org/plosone/s/licenses-and-copyright.

 4.1. You may seek permission from the original copyright holder of Figure 1 to publish the content specifically under the CC BY 4.0 license. 

 4.2. If you are unable to obtain permission from the original copyright holder to publish these figures under the CC BY 4.0 license or if the copyright holder’s requirements are incompatible with the CC BY 4.0 license, please either i) remove the figure or ii) supply a replacement figure that complies with the CC BY 4.0 license. Please check copyright information on all replacement figures and update the figure caption with source information. If applicable, please specify in the figure caption text when a figure is similar but not identical to the original image and is therefore for illustrative purposes only.

Response:

We thank the academic editor for his announcements. We have replaced Figure 1 with an image for public use, specifically from CIA maps at domain:

https://www.cia.gov/library/publications/the-world-factbook/index.html

Comment 5: We note you have included a table to which you do not refer in the text of your manuscript. Please ensure that you refer to Table 9, 10, 11 in your text; if accepted, production will need this reference to link the reader to the Table.

Response:

We agree with you. We have corrected these errors. All changes have been marked in red color in the revised manuscript.  

II. Responses to Reviewer #1

General comment: I have read the paper entitled "Design Deep Neural Network Architecture using a Genetic Algorithm for Estimation of Pile Bearing Capacity". In essence, the paper suggests a deep ANN-based predictive model for pile bearing capacity. It is interesting that authors used GA for reducing the number of features from 10 to 4. The paper is well written and well organized. Although compared to the previous publications, slight contribution was observed, presenting new sets of real data is always of interest as it can constitute common sense.

Response:

We thank Reviewer #1 for his nice and constructive comments, which help us in improving the quality of our work. 

Comment 1: Hence, firstly authors are requested to present at least 100 sets of data in the appendix 

Response: Yes, we do agree and already added our data as request in the appendix.

Comment 2

Response: 

We have already used 4 performance indicators in our study: R2, MAE, RMSE and IA. We believe that it is appropriate to use the VAF index more, but in this study, since many other indicators have been used, we will pay attention to the use of VAF in the upcoming studies.

Comment 3: Enhance the literature review considerably by providing a Table of previous AI-based works in the field of foundation engineering including deep foundation, shallow foundation, thin-walled foundations below are some recommendations however authors do not have to cite them necessarily if they find them irrelevant. the implemented soft computing technique, type of foundations, dataset size, R or R2 should be highlighted in this table.

Response: 

We agree with reviewer comments. We found that supplementing the previous research results was the right thing to do, in particular, we added Table 12 to the manuscript. The contents have been added in the revised manuscript as follow:

Table 12. Comparison with other studies

Author Model Foundation type Number of samples R2 RMSE

Momeni el al. [1]

ANFIS Thin-walls 150 0.875 0.048

 ANN 0.71 0.529

Momeni el al. [2]

GPR Piles 296 0.84 -

Kulkarni el al. [3]

GA-ANN Rock-socketed piles 132 0.86 0.0093

Jahed Armaghani el al. [4] 

ANN 0.808 0.135

 PSO-ANN 0.918 0.063

The present study GA-DNN Piles 472 0.882 109.965

Table 12 presents some research results on ML applications in foundation engineering. The results of this study as well as previous studies show that the expected foundation effectiveness of ML technique in foundation engineering with prediction results of foundation load is mostly reaching R2 from 0.8 to 0.9.

Comment 4: It should be clearly highlighted in the introduction that in what aspect the presented paper is different from other studies (like implementation of deep learning)

Response: 

We agree with reviewer comments. We have already added section 2. Significance of the research study into our revise manuscript. The contents have been added in the revised manuscript as follow:

The numerical or experimental methods in the existing literature still have some limitations, such as lack of data set samples (Marto et al. [55] with 40 samples; Momeni et al. [45] with 36 samples; Momeni et al.[56] with 150 samples; Bagińska and Srokosz [57] with 50 samples; Teh et al. [58] with 37 samples), refinement of ML approaches or failure to fully consider key parameters which affects the predicting results of the model.

For this, the contribution of the present work can be marked through the following ideas: (i) large data set, including 472 experimental tests; (ii) reduce the input variables from 10 to 4 which help the model achieve more accurate results with faster training time, (iii) automatically design the optimal architecture for the DLNN model, all key parameters are considered, include: the number of hidden layers, the number of neurons in each hidden layer, the activation function and the training algorithm. In which, the number of hidden layers is not fixed but can be selected through cross-mating between the parent with different chromosome length. Besides, the randomness in the order of the training data set is also considered to assess the stability of predicting result of models with the training, validate and testing set.

Comment 5: Despite AI advantages, limitations of these methods should be clearly highlighted.

Response 

We thank you for this very interesting comment. We have already added some limitations of machine learning method to our revise script. The contents have been added in the revised manuscript as follow:

Despite the recent successes of machine learning, this method has some limitations to keep in mind: It requires large amounts of of hand-crafted, structured training data and cannot be learned in real time. In addition, ML models still lack the ability to generalize conditions other than those encountered during the training. Therefore, the ML model only correctly predicts in a certain data range but is not generalized in all cases.

Comment 6: A competitor like conventional BP-ANN is needed for comparison purposes or the prediction performance of the proposed AI-based predictive model should be checked against other works.

Response: 

Thanks for your interesting comment. Back Propagation (BP) is a gradient descent optimization algorithm and commonly used to find out optimal weights and biases of neural networks through training. Of course, the Genetic Algorithm is one of the optimal algorithms that does not use gradient descent, however, in this study, we did not use the genetic algorithm in the training model but use it to self to automate choice of network architecture (include as follows: number of hidden layers, number of neurons in each hidden layer, training algorithm and activation function for hidden neurons). Therefore, we don’t believe that it is make sense when compare GA - DLNN model with BP - ANN (BP - DLNN) model.

Comment 7: Checking the English is suggested.

Response: Thanks, we would check carefully.

 

III. Responses to Reviewer #2

General comment: Introduction: As there are plenty of studies involving the GA optimized DNNs in this field I strongly advise to explain the novelty clearly and justify the need for this particular research.

Response: 

We agree with reviewer comments. We have already added section 2. Significance of the research study into our revise manuscript. The contents have been added in the revised manuscript as follow:

The numerical or experimental methods in the existing literature still have some limitations, such as lack of data set samples (Marto et al. [55] with 40 samples; Momeni et al. [45] with 36 samples; Momeni et al. [56] with 150 samples; Bagińska and Srokosz [57] with 50 samples; Teh et al. [58] with 37 samples), refinement of ML approaches or failure to fully consider key parameters which affects the predicting results of the model.

For this, the contribution of the present work can be marked through the following ideas: (i) large data set, including 472 experimental tests; (ii) reduce the input variables from 10 to 4 which help the model achieve more accurate results with faster training time, (iii) automatically design the optimal architecture for the DLNN model, all key parameters are considered, include: the number of hidden layers, the number of neurons in each hidden layer, the activation function and the training algorithm. In which, the number of hidden layers is not fixed but can be selected through cross-mating between the parent with different chromosome length. Besides, the randomness in the order of the training data set is also considered to assess the stability of predicting result of models with the training, validate and testing set.

Comment 1: Section 2.2. Data preparation: line 2 "[...] all the factors affecting the pile bearing capacity were considered.". I suggest to put it that way "all the known factors" as all the factors affecting the bearing capacity might not be discovered yet.

Response: 

Thank you for your comment. We found it to be very helpful.

Comment 2: Section 4.2. Optimization of DLNN Architecture: line 10 "[...] model performed well better performance"?

Response: 

Thank you for your comment. That was a mistake and we fixed it in revise manuscript.

Comment 3: Section 4.3. Predictive Capability of the Models: In Tab. 7. you compare the "predictive capability of the models" on three datasets (training, validation and testing). Low error achieved on the training and validation dataset does not mean that the model will predict accurately (i.e. perform good on testing set). When the function fits the training data very well the model's predictions can often be not so accurate (over fitting), cause the model has lower generalization ability. Therefore, the predictive capability of the model can only be measured with the error obtained on the testing dataset.

Response: 

Thank you for your comment. I agree with your ideas, the predictive capability of the model can only be measured with the error obtained on the testing dataset and when the model too fits with training data, it’s might cause over fitting. Amongst our three DLNN models, the 4-in-DA-DLNN model gives the best results on the training and validation set and the test set. In my opinion, this model is not too over fitting, however, in upcoming studies, we will try to adjust the model carefully to get more desired results on testing set to get a more general model.

Comment 4: Conclusions: I suggest pointing out the main achievement of this study, maybe mentioning possible applications of the developed model and future research perspectives.

Response:

 Thank you for your comment. We also have already added this content section 2.

IV. Responses to Reviewer #3

General comment: The manuscript describes a technically sound scientific study with data that supports the conclusions. The experiments were performed rigorously with appropriate controls (four control conditions: R2, IA, RMSE and MAE), replication and sample size (1000 replicates, 3 structure configurations). On the basis of the obtained data, appropriate conclusions were drawn. The statistical analysis was performed appropriately and rigorously, although the presentation of the results shows a lack of consistency in the data:

Response: 

We thank Reviewer #3 for his nice and constructive comments, which help us in improving the quality of our work. 

Comment 1: Figure 11 shows the R values whose R2 equivalents do not match the values summarized in Table 7.

Response: Thank you for your comment. That was a mistake and we fixed it in revise manuscript.

Comment 2: The results of the analyzes presented in Tables 9, 10 and 11 are identical and should contain collected values according to different three criteria.

The authors provided graphical access to all the data underlying the findings in their manuscript. The manuscript is clearly presented and written in standard English, but there are some minor typing errors in the text, e.g.:

Response: Thank you for your comment. That was a mistake and we fixed it in revise manuscript.

Comment 3: page 12: “The initialization parameters of GA used in this study are given in Tables 3.” (should be: “… in Table 3.”).

Response: Thank you for your comment. That was a mistake and we fixed it in revise manuscript.

Comment 4: page 21: “On the validation data set, the 4-input GA-DLNN model gave similar results to the 10-input GA-DLNN model and outperformed the 4-input GA-DLNN model with satisfactory accuracy…” (should be: “… the 4-input DLNN model with satisfactory accuracy…”).

Response: Thank you for your comment. That was a mistake and we fixed it in revise manuscript.

Comment 5: The authors of the manuscript made every effort to ensure that the final version of the text was of the highest possible scientific and editorial level, but the reviewer believes that the combination of graphs presented in Figures 9 and 10 would allow for an easier comparative analysis of the results. The aim of the graphical presentation of the results in Figure 12 was a comparative analysis - according to the reviewer, changing the vertical scale (starting from higher values) will allow to emphasize the differences between the examined structures and the types of tests performed.

Response: 

Thank you for your comment. We found it very helpful and combined the Figure 9 and 10.

(1) Please ensure that you refer to Figure 4 in your text as, if accepted, production will need this reference to link the reader to the figure.

 Figure 4 is linked in text of the manuscript.

References

[1] E. Momeni, D. J. Armaghani, S. A. Fatemi, and R. Nazir, “Prediction of bearing capacity of thin-walled foundation: a simulation approach,” Engineering with Computers, vol. 34, no. 2, pp. 319–327, Apr. 2018, doi: 10.1007/s00366-017-0542-x.

[2] E. Momeni, M. B. Dowlatshahi, F. Omidinasab, H. Maizir, and D. J. Armaghani, “Gaussian Process Regression Technique to Estimate the Pile Bearing Capacity,” Arab J Sci Eng, vol. 45, no. 10, pp. 8255–8267, Oct. 2020, doi: 10.1007/s13369-020-04683-4.

[3] R. U. Kulkarni, D. M. Dewaikar, and Indian Institute of Technology Bombay, “Prediction of Interpreted Failure Loads of Rock-Socketed Piles in Mumbai Region using Hybrid Artificial Neural Networks with Genetic Algorithm,” IJERT, vol. V6, no. 06, Art. no. 06, Jun. 2017, doi: 10.17577/IJERTV6IS060196.

[4] D. Jahed Armaghani, R. S. N. S. B. R. Shoib, K. Faizi, and A. S. A. Rashid, “Developing a hybrid PSO–ANN model for estimating the ultimate bearing capacity of rock-socketed piles,” Neural Comput & Applic, vol. 28, no. 2, pp. 391–405, Feb. 2017, doi: 10.1007/s00521-015-2072-z.

---

## [Decision Letter · Decision Letter 1]

16 Nov 2020

Design Deep Neural Network Architecture using a Genetic Algorithm for Estimation of Pile Bearing Capacity

PONE-D-20-26359R1

Dear Dr. Pham,

We’re pleased to inform you that your manuscript has been judged scientifically suitable for publication and will be formally accepted for publication once it meets all outstanding technical requirements.

Kind regards,

**Le Hoang Son, Ph.D**

Academic Editor

PLOS ONE

**Comments to the Author**

1. If the authors have adequately addressed your comments raised in a previous round of review and you feel that this manuscript is now acceptable for publication, you may indicate that here to bypass the “Comments to the Author” section, enter your conflict of interest statement in the “Confidential to Editor” section, and submit your "Accept" recommendation.

Reviewer #1: All comments have been addressed

Reviewer #2: All comments have been addressed

Reviewer #3: All comments have been addressed

2. Is the manuscript technically sound, and do the data support the conclusions?

Reviewer #1: Yes

Reviewer #2: Yes

Reviewer #3: Yes

3. Has the statistical analysis been performed appropriately and rigorously? 

Reviewer #1: Yes

Reviewer #2: Yes

Reviewer #3: Yes

4. Have the authors made all data underlying the findings in their manuscript fully available?

Reviewer #1: Yes

Reviewer #2: Yes

Reviewer #3: Yes

5. Is the manuscript presented in an intelligible fashion and written in standard English?

Reviewer #1: Yes

Reviewer #2: Yes

Reviewer #3: Yes

6. Review Comments to the Author

**Reviewer #1**: Authors have addressed all the comments properly and the paper can be accepted. The paper is now more interesting. The significance as well as the limitation of the work is now highlighted. The literature review is enhanced and the result of this study is compared with other relevant works.

**Reviewer #2**: (No Response)

**Reviewer #3**: The revised version of the work meets the reviewer's expectations and will certainly find great interest among readers dealing with this type of research.

The reviewer believes that minor editorial errors (such as table numbering) will be removed during the publication process.

---

## [Editor Report · Acceptance letter]

3 Dec 2020

PONE-D-20-26359R1 

Design deep neural network architecture using a genetic algorithm for estimation of pile bearing capacity 

Dear Dr. Pham:

I'm pleased to inform you that your manuscript has been deemed suitable for publication in PLOS ONE. Congratulations! Your manuscript is now with our production department. 

Kind regards, 

on behalf of

Prof. Le Hoang Son 

Academic Editor

PLOS ONE